# Weighted Cumulative Past Extropy and Its Inference

**DOI:** 10.3390/e24101444

**Published:** 2022-10-11

**Authors:** Mohammad Reza Kazemi, Majid Hashempour, Maria Longobardi

**Affiliations:** 1Department of Statistics, Faculty of Science, Fasa University, Fasa IS74, Iran; 2Department of Statistics, University of Hormozgan, Bandar Abbas 79177, Iran; 3Dipartimento di Biologia, Università degli Studi di Napoli Federico II, 80126 Naples, Italy

**Keywords:** weighted cumulative past extropy, reliability theory, empirical extropy, goodness of fit test, 62B10, 62N05

## Abstract

This paper introduces and studies a new generalization of cumulative past extropy called weighted cumulative past extropy (WCPJ) for continuous random variables. We explore the following: if the WCPJs of the last order statistic are equal for two distributions, then these two distributions will be equal. We examine some properties of the WCPJ, and a number of inequalities involving bounds for WCPJ are obtained. Studies related to reliability theory are discussed. Finally, the empirical version of the WCPJ is considered, and a test statistic is proposed. The critical cutoff points of the test statistic are computed numerically. Then, the power of this test is compared to a number of alternative approaches. In some situations, its power is superior to the rest, and in some other settings, it is somewhat weaker than the others. The simulation study shows that the use of this test statistic can be satisfactory with due attention to its simple form and the rich information content behind it.

## 1. Introduction

In recent years, there has been strong interest in the measurement of the uncertainty of probability distributions, which is called entropy. The probabilistic concept of entropy was developed by [1]. For an absolutely continuous random variable *X*, the Shannon entropy is defined as
$$H\left(X\right)=-E\left(\log f\left(X\right)\right)=-\int_{-\infty }^{+\infty }f\left(x\right)\log f\left(x\right)dx,$$
where “log” means the natural logarithm, and f(x) is the probability density function (pdf) of a random variable *X*. Several applications of entropy in information theory, economics, communication theory, and physics are well developed in the literature, (see Cover and Thomas, [2]). Belis and Guiasu [3] and Guiasu [4] considered a weighted entropy measure as
(1)$$H^{w}\left(X\right)=-E\left(X\log f\left(X\right)\right)=-\int_{-\infty }^{+\infty }xf\left(x\right)\log f\left(x\right)dx,$$
where by assigning greater importance to larger values of *X*, the weight *x* in (Equation 1) emphasizes the occurrence of the event X=x. Reference [5] stated the necessity of the existence of the weighted measures of uncertainty. In the Shanon entropy H(X), only the pdf of the random variable *X* is regarded. Moreover, it is known that this information measure is shift-independent, in the sense that the information content of a random variable *X* is equal to that of X+b. Indeed, some applied fields such as neurobiology do not tend to deal with shift-independent but shift-dependent. Further research was conducted to generalize the concept of entropy, for example, by replacing the pdf f(x) with the survival function $\bar{F}\left(x\right)$, ref. [6] introduced the cumulative residual entropy (CRE) as
(2)$$\begin{array}{c} E\left(X\right)=-\int_{0}^{+\infty }\bar{F}\left(x\right)\log\bar{F}\left(x\right)dx. \end{array}$$
Moreover, continued generalizations recently include a more tractable measure of information, which is the dual of entropy, called extropy, introduced by [7], which has the following form:(3)$$\begin{array}{ccc} J(X) & = & -\frac{1}{2}\int_{0}^{+\infty }f^{2}\left(x\right)dx \end{array}$$(4)$$\begin{array}{ccc} & = & -\frac{1}{2}\int_{0}^{1}f\left(F^{-1}\left(u\right)\right)du. \end{array}$$
After that, a number of researchers have worked to identify the behavior of this concept in some complex schemes. In fact, both entropy and extropy provide us the information content associated with the random variable *X*. As stated before, the extropy is a measure of uncertainty introduced as dual to the entropy. The most important advantage of extropy is that it is easy to compute. References [8,9,10] characterized the behavior of extropy and its generalization in record values, order statistics schemes, and mixed systems, respectively. Moreover, the extropy properties of the ranked set sampling were given in [11]. In addition to the work of [12] in which the concept of extropy was generalized to cumulative residual extropy, reference [13] investigated the properties of this term in both theoretical and applied aspects based on a version of the ranked set sampling. Moreover, Vaselabi et al., Buono and Longobardi, Kazemi et al. [14,15,16] considered varextropy, Deng extropy, and fractional Deng extropy as generalizations of extropy. Furthermore, References [17,18,19] considered dynamic weighted extropy, the extropy of past lifetime distribution, and the extropy of *k*-records, respectively. For the problem of estimation and inference of the extropy, one can see, for example [20,21], and so on. An alternative measure of the uncertainty of a random variable *X* called cumulative residual extropy (CRJ) was introduced by [12] as
(5)$$\begin{array}{ccc} EJ(X) & = & -\frac{1}{2}\int_{0}^{+\infty }{\bar{F}}^{2}\left(x\right)dx \end{array}$$
(6)$$\begin{array}{ccc} & = & -\frac{1}{2}\int_{0}^{1}\frac{u^{2}du}{f\left(F^{-1}\left(1-u\right)\right)}. \end{array}$$
As we see, this measure is a generalization of the so-called extropy of [7] in which the survival function $\bar{F}\left(x\right)$ plays the role of the pdf f(x) in (Equation 3). Since the pdf f(x) is the derivative of the cumulative distribution function F(x) (cdf), cdf is more convenient to work with. Therefore, because of its convenience, some researchers prefer to work with CRJ than extropy. In the following, the basic idea is to replace the pdf with the cdf in the extropy definition (Equation 3). The cdf is more regular than the pdf, because the pdf is computed as the derivative of the cdf. For the dual measure for a random variable *X*, we can define the cumulative past extropy (CPJ) as
(7)$$\begin{array}{ccc} \bar{E}J\left(X\right) & = & -\frac{1}{2}\int_{0}^{+\infty }F^{2}\left(x\right)dx \end{array}$$
(8)$$\begin{array}{ccc} & = & -\frac{1}{2}\int_{0}^{1}\frac{u^{2}du}{f\left(F^{-1}\left(u\right)\right)}. \end{array}$$
The $\bar{E}J\left(X\right)$ is suitable to measure information when uncertainty is related to the past, and the empirical version of the CPJ can be easily obtained rather than the empirical version of the extropy itself. So, one can explore the applications of the CPJ in providing inferential methods. It is reasonable to define the CPJ only for random variables with bounded support, since this measure will be equal to −∞ for all random variables with unbounded support. The rest of this paper is organized as follows. In Section 2, we introduce the weighted cumulative past extropy as well as analyzing some of its properties, and some examples are presented. Section 3 considers the WCPJ of order statistics. Furthermore, we explore that when WCPJs of the last order statistic are equal for two distributions, these two distributions will be equal. In Section 4, some bounds and inequalities are achieved. Section 5 focuses on certain connections to reliability theory. Finally, in Section 6, an empirical version of the WCPJ is provided, and a hypothesis testing problem is carried out for a goodness of fit test of the standard uniform distribution.

## 2. Weighted Cumulative Past Extropy

In this section, we introduce a new information measure called weighted cumulative past extropy (WCPJ). The cumulative past extropy can be generalized to weighted cumulative past extropy. The main objective of the study is to extend weighted extropy to random variables with continuous distributions.

**Definition** **1.**
*Let X be a nonnegative absolutely continuous random variable having cdf F(x). We define the WCPJ of X by*

(9)
$$\begin{array}{ccc} {\bar{E}}^{w}J\left(X\right) & = & -\frac{1}{2}\int_{0}^{+\infty }xF^{2}\left(x\right)dx \end{array}$$


(10)
$$\begin{array}{ccc} & = & -\frac{1}{2}\int_{0}^{1}\frac{u^{2}F^{-1}\left(u\right)}{f\left(F^{-1}\left(u\right)\right)}du. \end{array}$$



The following equality can be used in the sequel.
(11)$$\begin{array}{c} {\bar{E}}^{w}J\left(X\right)=-\frac{1}{2}\int_{0}^{\infty }\int_{y}^{\infty }F_{X}^{2}\left(x\right)dxdy. \end{array}$$
As stated in the introduction, similar to the CPJ, the value of the WCPJ is −∞ for all random variables with unbounded support. So, our definition for WCPJ should be restricted to all random variables with bounded support. Let *X* be a nonnegative random variable with bounded support *S*; then, the WCPJ of *X* is defined as
(12)$${\bar{E}}^{w}J\left(X\right)=-\frac{1}{2}\int_{0}^{\sup S}xF^{2}\left(x\right)dx.$$
Now, we evaluate the WCPJ of some distributions.

**Example** **1.**
*Let X have the power distribution with the cdf, $F\left(x\right)={\left(\frac{x}{\beta }\right)}^{\theta }$, x∈(0,β), θ>0. Then,*

(13)
$$\bar{E}J\left(X\right)=-\frac{\beta }{2(2\theta +1)},$$

*and*

(14)
$${\bar{E}}^{w}J\left(X\right)=-\frac{\beta^{2}}{4\left(\theta +1\right)}.$$

*We conclude ${\bar{E}}^{w}J\left(X\right)=\frac{2\theta +1}{2(\theta +1)}\beta \bar{E}J\left(X\right)$. If $\beta >\frac{2(\theta +1)}{2\theta +1}$, then ${\bar{E}}^{w}J\left(X\right)>\bar{E}J\left(X\right)$, and if $\beta <\frac{2(\theta +1)}{2\theta +1}$, then ${\bar{E}}^{w}J\left(X\right)<\bar{E}J\left(X\right)$.*


**Example** **2.**
*Let X be a uniform random variable such that X∼U(a,b). Then,*

$$\bar{E}J\left(X\right)=\frac{a-b}{6},$$

*and*

$$\begin{array}{c} {\bar{E}}^{w}J\left(X\right)=-\frac{b-a}{24}\left(a+3b\right). \end{array}$$

*We conclude*

$$\begin{array}{c} {\bar{E}}^{w}J\left(X\right)=\left(\frac{3b+a}{4}\right)\bar{E}J\left(X\right)=\left(\frac{E(X)+b}{2}\right)\bar{E}J\left(X\right). \end{array}$$

*If E(X)>2−b, then ${\bar{E}}^{w}J\left(X\right)>\bar{E}J\left(X\right)$, and if E(X)<2−b, then ${\bar{E}}^{w}J\left(X\right)<\bar{E}J\left(X\right)$.*


In the following, the effect of the linear transformation on the WCPJ will be studied.

**Proposition** **1.**
*Let X be a nonnegative random variable. If Y=aX+b, a>0,b≥0, then*

(15)
$$\begin{array}{c} {\bar{E}}^{w}J\left(Y\right)=a^{2}{\bar{E}}^{w}J\left(X\right)+ab\bar{E}J\left(X\right). \end{array}$$



**Theorem** **1.**
*Let X be a nonnegative continuous random variable with bounded support S.Then, we have*
*(i)* 

${\bar{E}}^{w}J\left(X\right)=-\frac{1}{2}E\left[H_{F}\left(\underline{X}\right)\right].$

*(ii)* 

${\bar{E}}^{w}J\left(X\right)=-\frac{1}{2}\left(E\left[H_{F}\right]-E\left[H_{F}\left(\bar{X}\right)\right]\right),$


*where $H_{F}\left(\bar{t}\right)=\int_{0}^{t}xF\left(x\right)dx$, $H_{F}\left(\underline{t}\right)=\int_{t}^{\sup S}xF\left(x\right)dx$, and $H_{F}=\int_{0}^{\sup S}xF\left(x\right)dx$.*



**Proof.** From Equation (Equation 9) and by Fubini’s theorem, we have
(16)$$\begin{array}{ccc} {\bar{E}}^{w}J\left(X\right) & = & -\frac{1}{2}\int_{0}^{\sup S}xF^{2}\left(x\right)dx=-\frac{1}{2}\int_{0}^{\sup S}xF\left(x\right)\int_{0}^{x}f\left(t\right)dtdx\\ & = & -\frac{1}{2}\int_{0}^{\sup S}f\left(t\right)\int_{t}^{\sup S}xF\left(x\right)dxdt\\ & = & -\frac{1}{2}E\left(\int_{X}^{\sup S}xF\left(x\right)dx\right). \end{array}$$
On the other hand,
(17)$$\int_{t}^{\sup S}xF\left(x\right)dx=\int_{0}^{\sup S}xF\left(x\right)dx-\int_{0}^{t}xF\left(x\right)dx.$$
The proof of part (ii) then follows from the substitution of (Equation 17) in (Equation 16). □

In the following, we express an upper bound of the WCPJ in terms of the extropy.

**Theorem** **2.**
*Let J(X) be the extropy of the random variable X and f(x)≤1 for all n; then,*

(18)
$${\bar{E}}^{w}J\left(X\right)\leq D^{*}\exp\left\{2J\left(X\right)\right\},$$

*where $D^{*}=-\frac{1}{2}\exp\left\{E\left[\log\left(XF^{2}\left(X\right)\right)\right]\right\}.$*


**Proof.** The proof is similar to that of Theorem 2.3 in [22]. □

**Remark** **1.**
*For a nonnegative and absolutely continuous random variable X with bounded support S, the weighted cumulative past extropy is nonpositive.*


## 3. Some Characterization Results Based on the Order Statistics

In this section, for some characterization results, the following lemma is needed.

**Lemma** **1.**
*Let g be a continuous function with support [0,1], such that $\int_{0}^{1}g\left(y\right)y^{m}dy=0$, for m≥0; then, g(y)=0, for all y∈[0,1].*


In the following, we provide the WCPJ of the last and first order statistics. As before, we assume that the random variable *X* has bounded support *S*. The WCPJ of the last order statistic is
(19)$${\bar{E}}^{w}J\left(X_{n:n}\right)=-\frac{1}{2}\int_{0}^{\sup S}xF_{X_{n:n}}^{2}\left(x\right)dx=-\frac{1}{2}\int_{0}^{\sup S}xF_{X}^{2n}\left(x\right)dx,$$
With a change in variable, $u=F_{X}\left(x\right)$, we are able to write
(20)$${\bar{E}}^{w}J\left(X_{n:n}\right)=-\frac{1}{2}\int_{0}^{1}\frac{u^{2n}F^{-1}\left(u\right)}{f(F^{-1}\left(u\right))}du.$$
Moreover, by using $F_{X_{1:n}}\left(x\right)=1-{\bar{F}}_{X}^{n}\left(x\right)$, we have
(21)$${\bar{E}}^{w}J\left(X_{1:n}\right)=-\frac{1}{2}\int_{0}^{\sup S}x{\left(1-{\bar{F}}^{n}\left(x\right)\right)}^{2}dx,$$
with another change in variable, $u=\bar{F}\left(x\right)$ in (Equation 21), we have
(22)$${\bar{E}}^{w}J\left(X_{1:n}\right)=-\frac{1}{2}\int_{0}^{1}\frac{{\left(1-u\right)}^{2n}F^{-1}\left(1-u\right)}{f(F^{-1}\left(1-u\right))}du.$$

**Remark** **2.**
*Let $\Lambda^{*}={\bar{E}}^{w}J\left(X_{n:n}\right)-{\bar{E}}^{w}J\left(X\right).$ Since $\Lambda^{*}>0$, the uncertainty of $X_{n:n}$ is more than that of X, for all n. If n=1, then ${\bar{E}}^{w}J\left(X_{n:n}\right)={\bar{E}}^{w}J\left(X\right).$*


Now, we evaluate the WCPJ of $X_{n:n}$ for some distributions.

**Example** **3.**
*Let X have a Power distribution with the cdf $F\left(x\right)={\left(\frac{x}{\beta }\right)}^{\theta }$, 0<x<θ, 0<θ. Then, $\bar{E}J\left(X\right)=-\frac{\beta }{2(2\theta +1)}$, ${\bar{E}}^{w}J\left(X\right)=-\frac{\beta^{2}}{4(\theta +1)}$, $\bar{E}J\left(X_{n:n}\right)=-\frac{\beta }{2(2n\theta +1)}$, and $EJ\left(X_{1:n}\right)=-\frac{\beta^{2}}{4(n\theta +1)}$. In the sequel, $\bar{E}J\left(X_{n:n}\right)=\frac{\theta +1}{n\theta +1}{\bar{E}}^{w}J\left(X\right)$.*


**Example** **4.**
*Assume that X has a uniform distribution with support on (a,b). Then, $\bar{E}J\left(X_{n:n}\right)=-\frac{b-a}{2(2n+1)}$, ${\bar{E}}^{w}J\left(X_{n:n}\right)=-\frac{b-a}{2(2n+1)}\left(b-\frac{b-a}{2(n+1)}\right)$, $\bar{E}J\left(X\right)=\frac{a-b}{6}$, and ${\bar{E}}^{w}J\left(X\right)=-\frac{b-a}{6}\left(3a+b\right)$.*


**Theorem** **3.**
*Let $X_{1},\cdots ,X_{n}$ and $Y_{1},\cdots ,Y_{n}$ be random samples from nonnegative continuous cdfs F(x) and G(x) and pdfs f(x) and g(x), respectively, with a common bounded support. Then, F(x)=G(x) if and only if ${\bar{E}}^{w}J\left(X_{n:n}\right)={\bar{E}}^{w}J\left(Y_{n:n}\right)$, for all n.*


**Proof.** The necessity is trivial. Therefore, it remains to prove the sufficiency part. If ${\bar{E}}^{w}J\left(X_{n:n}\right)={\bar{E}}^{w}J\left(Y_{n:n}\right)$, for all *n*, then we have
$$\begin{array}{c} -\frac{1}{2}\int_{0}^{1}u^{2n}\left[\frac{F^{-1}\left(u\right)}{f(F^{-1}\left(u\right))}-\frac{G^{-1}\left(u\right)}{g(G^{-1}\left(u\right))}\right]du=0. \end{array}$$
By using Lemma 1, we obtain
$$\begin{array}{c} \frac{F^{-1}\left(u\right)}{f(F^{-1}\left(u\right))}=\frac{G^{-1}\left(u\right)}{g(G^{-1}\left(u\right))}. \end{array}$$
In the following, we have $F^{-1}\left(u\right)dF^{-1}\left(u\right)/du=G^{-1}\left(u\right)dG^{-1}\left(u\right)/du$, u∈[0,1]. Since $dF^{-1}\left(u\right)/du=1/f\left(F^{-1}\left(u\right)\right)$, it will be concluded that $F^{-1}\left(u\right)=G^{-1}\left(u\right)$, u∈[0,1]. □

**Theorem** **4.**
*Suppose that $X_{1},\cdots ,X_{n}$ and $Y_{1},\cdots ,Y_{n}$ are random samples from nonnegative continuous cdfs F(x) and G(x) and pdfs f(x) and g(x), respectively, such that $F\left(x^{*}\right)=G\left(x^{*}\right)$. Then, F(x)=G(x), for $x<x^{*}$, if and only if*

(23)
$${\bar{E}}^{w}J\left(X_{j:n}|X_{j+1:n}=x^{*}\right)={\bar{E}}^{w}J\left(Y_{j:n}|Y_{j+1:n}=x^{*}\right).$$



**Proof.** Suppose ${\bar{E}}^{w}J\left(X_{j:n}|X_{j+1:n}=x^{*}\right)={\bar{E}}^{w}J\left(Y_{j:n}|Y_{j+1:n}=x^{*}\right)$; that is, the WCPJ of the last order statistic for two distributions F(x) and G(x) truncated at $x^{*}$ are equal. Thus, by Theorem 3, these two truncated distributions are equal, which leads to F(x)=G(x) for $x<x^{*}$.Conversely, if F(x)=G(x) for $x<x^{*}$, then by assumption, $F\left(x^{*}\right)=G\left(x^{*}\right)$, and F(x) and G(x) truncated at $x^{*}$ are equal for $x<x^{*}$; that is,
$$\frac{F\left(x^{*}\right)-F\left(x\right)}{1-F(x^{*})}=\frac{G\left(x^{*}\right)-G\left(x\right)}{1-G(x^{*})},\;\;x<x^{*}.$$
The distribution of $X_{j:n}$, given that $X_{j+1:n}=x^{*}$, is the same as the distribution of the last order statistic obtained from a sample of size n−j−1 from a population whose distribution F(x) is truncated at $x^{*}$. For more details, see [23]. By Theorem 3, we conclude that ${\bar{E}}^{w}J\left(X_{j:n}|X_{j+1:n}=x^{*}\right)={\bar{E}}^{w}J\left(Y_{j:n}|Y_{j+1:n}=x^{*}\right)$. □

## 4. Some Inequalities

In this section, we obtain some upper and lower bounds for the WCPJ.

**Proposition** **2.**
*Let X be a nonnegative continuous random variable with the cdf $F_{X}\left(x\right)$ and bounded support S=[k,supS). Then, we obtain*

(24)
$${\bar{E}}^{w}J\left(X\right)\leq k\;\bar{E}J\left(X\right).$$



**Corollary** **1.**
*Let X be a continuous random variable with the cdf F(x) and support [0,k]. Then,*
*(i)* 

$k\bar{E}J\left(X\right)\leq {\bar{E}}^{w}J\left(X\right).$

*(ii)* 
*${\bar{E}}^{w}J\left(X\right)\leq -\frac{H_{F}\left(\bar{k}\right)}{2}\log\left[1+\left(\frac{2H_{F}\left(\bar{k}\right)}{k^{2}}\right)\right]$,*

*where $H_{F}\left(\bar{k}\right)=\int_{0}^{k}xF\left(x\right)dx$.*



In the following, stochastic orders of two distributions in terms of their characteristics are considered. For more details, one can see [24]. In the sequel, we show that the ordering of the WCPJ is implied by the usual stochastic order.

**Definition** **2.**
*A random variable $X_{1}$ is said to be smaller than $X_{2}$ in the usual stochastic order, denoted by $X_{1}\leq_{st}X_{2}$, if $P\left(X_{1}\geq x\right)\leq P\left(X_{2}\geq x\right)$ for all x.*


**Definition** **3.**
*A random variable $X_{1}$ is said to be smaller than $X_{2}$ in the WCPJ order, denoted by $X_{1}\leq_{wcpj}X_{2}$, if*

(25)
$$\begin{array}{c} {\bar{E}}^{w}J\left(X_{1}\right)\leq {\bar{E}}^{w}J\left(X_{2}\right). \end{array}$$



**Proposition** **3.**
*Let $X_{1}$ and $X_{2}$ be nonnegative and continuous random variables. If $X_{1}\leq_{st}X_{2}$, then $X_{1}\leq_{wcpj}X_{2}$.*


**Example** **5.**
*Let X and Y be two random variables with the cdfs $F_{X}\left(x\right)=x,\;x\in \left[0,1\right]$ and $F_{Y}\left(x\right)=x^{2},\;x\in \left[0,1\right]$, respectively. It is seen that $X\leq_{st}Y$, and $X\leq_{wcpj}Y$.*


In the following, we find a lower bound for ${\bar{E}}^{w}J\left(X\right)$.

**Remark** **3.**
*Let X be a nonnegative random variable with the cdf F(x) and bounded support S. Then,*

(26)
$$\begin{array}{c} {\bar{E}}^{w}J\left(X\right)\geq \frac{1}{2}H_{F}\log\left(\frac{H_{F}}{A}\right)+K, \end{array}$$

*where $K=-\frac{1}{2}\int_{0}^{\sup S}x^{2}F^{2}\left(x\right)dx$ and $A=\int_{0}^{\sup S}F\left(x\right)dx$.*


**Proof.** By using the log-sum inequality, we obtain
(27)$$\begin{array}{c} -\int_{0}^{\sup S}xF\left(x\right)\log xdx\leq -H_{F}\log\left(\frac{H_{F}}{\int_{0}^{\sup S}F\left(x\right)dx}\right). \end{array}$$
Using ${\bar{F}}^{2}\left(x\right)\leq \bar{F}\left(x\right)$ and the inequality 1−x≤−logx for 0<x, we obtain
(28)$$\begin{array}{c} \int_{0}^{\sup S}x\left(1-x\right)F^{2}\left(x\right)dx\leq -H_{F}\log\left(\frac{H_{F}}{\int_{0}^{\sup S}F\left(x\right)dx}\right). \end{array}$$
By multiplying both sides of (Equation 28) by −1/2, we have
$$\begin{array}{c} {\bar{E}}^{w}J\left(X\right)\geq -\frac{1}{2}\left[\int_{0}^{\sup S}x^{2}F^{2}\left(x\right)dx-H_{F}\log\left(\frac{H_{F}}{\int_{0}^{\sup S}F\left(x\right)dx}\right)\right], \end{array}$$
*which completes the proof. □*

## 5. Connections to Reliability Theory

In this section, the connection between the WCPJ and reliability theory will be considered. The inactivity time function is of interest in many fields such as survival analysis, actuarial studies, economics, reliability, etc. The inactivity time is thus the duration of the time occurring between the inspection time *t* and the failure time *X*, given that at time *t* the system was found to be down. If *X* is the lifetime of a system, then the inactivity time of the system is denoted by $\left(t-X|X\leq t\right)$, t≥0. Let *X* be a nonnegative continuous random variable with the cdf F(x), such that E(X) is finite. The mean inactivity time (MIT) function of *X* is defined as
(29)$$MIT\left(t\right)=E\left(t-X|X\leq t\right)=\int_{0}^{t}\frac{F(x)}{F(t)}dx,\;t\geq 0.$$
This function has been used in various contexts of survival analysis and reliability theory involving characterization and stochastic orders of random lifetime. For more details, see [25,26,27,28,29,30]. In the following theorem, we prove that the WCPJ has a relation to the second moment of the inactivity time (SMIT) function.

**Definition** **4.**
*Let X be a nonnegative continuous random variable. Then, for all t≥0, we define the second moment of the inactivity time (SMIT) as*

(30)
$$\begin{array}{c} SMIT\left(t\right)=E\left({\left(t-X\right)}^{2}|X\leq t\right). \end{array}$$



It can be easily seen that
(31)$$\begin{array}{c} SMIT\left(t\right)=2tMIT\left(t\right)-\int_{0}^{t}2x\frac{F(x)}{F(t)}dx. \end{array}$$

**Theorem** **5.**
*Let X be a nonnegative continuous random variable with bounded support S, reversed hazard rate function rh(x), SMIT function, and weighted cumulative extropy ${\bar{E}}^{w}J\left(X\right)$. Thus,*

(32)
$${\bar{E}}^{w}J\left(X\right)\leq -\frac{1}{4}E\left[SMIT(X)\right]+C^{*},$$

*where $C^{*}=2^{-1}\left[E\left(X\cdot MIT\left(X\right)\right)-H_{F}\right].$*


**Proof.** $$\begin{array}{ccc} E(SMIT(X)) & = & 2E\left[X\cdot MIT\left(X\right)\right]-2\int_{0}^{\sup S}\int_{x}^{\sup S}x\;rh\left(t\right)F\left(x\right)dtdx\\ & = & 2E\left[X\cdot MIT\left(X\right)\right]-2\int_{0}^{\sup S}xF\left(x\right)\left|\log F\left(x\right)\right|dx\\ & \leq & 2E\left[X\cdot MIT\left(X\right)\right]-2\int_{0}^{\sup S}xF\left(x\right)dx+2\int_{0}^{\sup S}xF^{2}\left(x\right)dx\\ & = & 2E\left[X\cdot MIT\left(X\right)\right)]-2H_{F}-4{\bar{E}}^{w}J\left(X\right). \end{array}$$
In the sequel, we have
$$\begin{array}{ccc} {\bar{E}}^{w}J\left(X\right) & \leq & \frac{1}{4}\left[-E\left(SMIT\left(X\right)\right)+2E\left(X\cdot MIT\left(X\right)\right)-2E\left({\bar{H}}_{F}\left(X\right)\right)\right]\\ & = & -\frac{1}{4}E\left[SMIT(X)\right]+\frac{1}{2}\left[E\left(X\cdot MIT\left(X\right)\right)-H_{F}\right], \end{array}$$
and the proof is complete. □

Equation (Equation 32) is useful when we have some information about the SMIT or its behavior. An alternative expression to (Equation 32) can be given in terms of the hazard rate function. The hazard rate function of a random variable *X* with pdf f(x) and survival function $\bar{F}\left(x\right)$ is defined as $h\left(x\right)=f\left(x\right)/\bar{F}\left(x\right)$.

**Proposition** **4.**
*Let X be a nonnegative continuous random variable with bounded support S, hazard rate function h(x) and a finite WCPJ. Then,*

(33)
$${\bar{E}}^{w}J\left(X\right)\geq E\left(Q\left(X\right)\right),$$

*where $Q\left(t\right)=-\frac{1}{2}\int_{t}^{\sup S}x\left(\int_{0}^{x}h\left(u\right)du\right)dx$.*


**Proof.** 

$$\begin{array}{ccc} {\bar{E}}^{w}J\left(X\right) & = & -\frac{1}{2}\int_{0}^{\sup S}xF\left(x\right)\int_{0}^{x}f\left(t\right)dtdx\\ & = & -\frac{1}{2}\int_{0}^{\sup S}f\left(t\right)\int_{t}^{\sup S}xF\left(x\right)dxdt\\ & \geq & \frac{1}{2}\int_{0}^{\sup S}f\left(t\right)\int_{t}^{\sup S}x\log F\left(x\right)dxdt\\ & = & -\frac{1}{2}\int_{0}^{\sup S}f\left(t\right)\left(\int_{t}^{\sup S}x\int_{0}^{x}h\left(u\right)dudx\right)dt. \end{array}$$


*□*


## 6. Empirical WCPJ

In this part, an estimator of the WCPJ is constructed by means of the empirical WCPJ. Suppose that $X_{1},\cdots ,X_{n}$ is a nonnegative, continuous, independent, and identically distributed random sample from a population having the cdf F(x). By using the plug-in method, we define the empirical weighted cumulative past extropy as
$${\bar{E}}_{n}^{w}J\left(X\right)=-\frac{1}{2}\int_{0}^{+\infty }xF_{n}^{2}\left(x\right)dx,$$
where $F_{n}\left(x\right)$ is the empirical distribution function. Let $X_{\left(1\right)},X_{\left(2\right)},...,X_{\left(n\right)}$ be the ordered statistics corresponding to the underlying random sample. Then, ${\bar{E}}_{n}^{w}J\left(X\right)$ can be rewritten in the form of the ordered statistics
(34)$${\bar{E}}_{n}^{w}J\left(X\right)=-\frac{1}{4}\sum_{i=1}^{n-1}\left(X_{\left(i+1\right)}^{2}-X_{\left(i\right)}^{2}\right){\left(\frac{i}{n}\right)}^{2}.$$
In the following, we use ${\bar{E}}_{n}^{w}J\left(X\right)$ in (Equation 34) for testing the uniformity of the random sample $X_{1},\cdots ,X_{n}$. Before dealing with a test statistic, we give the following nice property of uniform distribution among all distributions defined on interval (0,1). For a random variable *X* with the cdf *F* and for $p\in \left(0,1\right)$, let $\psi_{p}J\left(F\right)$ be defined as
$$\psi_{p}J\left(F\right)=-\frac{1}{2}\int_{0}^{p}xF^{2}\left(x\right)dx.$$

It is trivial that for the uniform random variable *X* on interval (0,1) with the cdf $F_{0}\left(x\right)=x$, $\psi_{p}J\left(F_{0}\right)=-p^{4}/8$. Suppose that for a cdf *F* in the class of cdfs defined on interval (0,1),$\psi_{p}J\left(F\right)=-p^{4}/8$. This means that *F* and $F_{0}$ have the same measure based on $\psi_{p}J\left(\cdot \right)$, i.e., $\psi_{p}J\left(F\right)=\psi_{p}J\left(F_{0}\right)$. So, one can see that
$$\int_{0}^{p}x\left(F^{2}\left(x\right)-F_{0}^{2}\left(x\right)\right)dx=0,\;\forall p\in \left(0,1\right).$$

It is known that the $\left(0,p\right)$ generate the Borel σ-algebra of Ω=(0,1]. Therefore, one can write
$$\int_{B}x\left(F^{2}\left(x\right)-F_{0}^{2}\left(x\right)\right)dx=0,\;\forall B\subseteq \left(0,1\right].$$
So, $F\left(x\right)=F_{0}\left(x\right),\;$ almost everywhere is obtained. $\psi_{p}J\left(F\right)$ is uniquely determined by the uniform distribution in the sense that for some cdfs defined on $\left(0,1\right)$, they take a value lower than $-p^{4}/8$ and for some of them, they take higher than $-p^{4}/8,$ and only for the standard uniform distribution, we have $\psi_{p}J\left(F_{0}\right)=-p^{4}/8$.

### 6.1. Uniform Goodness of Fit Test

Based on this last property, a test statistic can be designed for the uniform goodness of fit test. One can construct a test statistic based on ${\bar{E}}_{n}^{w}J\left(X\right)$ in (Equation 34), which is the sampling counterpart of the WCPJ measure. For this goodness of fit test problem, we want to test whether the given random sample $X_{1},\cdots ,X_{n}$ is supported by the standard uniform distribution. In other words, we want to test a hypothesis testing $H_{0}:F=F_{0}$ against an alternative $H_{1}:F\neq F_{0}$, where $F_{0}$ is the cdf of the standard uniform distribution. A simple nonparametric test statistic is based on ${\bar{E}}_{n}^{w}J\left(X\right),$ as mentioned before. Indeed, ${\bar{E}}_{n}^{w}J\left(X\right)$ is our test statistic. In the next stage of our hypothesis testing, we should provide the critical region for the uniform goodness of fit test problem. The critical region is then obtained in the sense that ${\bar{E}}_{n}^{w}J\left(X\right)$ is less than or greater than two values $K_{1}\left(\alpha \right)$ and $K_{2}\left(\alpha \right)$, respectively, where α is a prespecified type I error rate; that is, one needs to determine $K_{1}\left(\alpha \right)$ and $K_{2}\left(\alpha \right)$, and whenever ${\bar{E}}_{n}^{w}J\left(X\right)<K_{1}\left(\alpha \right)$ or ${\bar{E}}_{n}^{w}J\left(X\right)>K_{2}\left(\alpha \right)$, then the null hypothesis of having a standard uniform distribution is rejected in favor of an alternative one. Since the distribution of the ${\bar{E}}_{n}^{w}J\left(X\right)$ is not easy to derive, then $K_{1}\left(\alpha \right)$ and $K_{2}\left(\alpha \right)$ can be estimated using the empirical quantile of the test statistic ${\bar{E}}_{n}^{w}J\left(X\right)$ under the standard uniform distribution. For a given type I error rate α and a large run number *N*, we generate a random sample $X_{1},\cdots ,X_{n}$ from the standard uniform distribution and then compute the value of ${\bar{E}}_{n}^{w}J\left(X\right).$ After that, we repeat this step for a large number of runs, i.e., N=100000. We sort these *N* values of ${\bar{E}}_{n}^{w}J\left(X\right).$ Then, $K_{1}\left(\alpha \right)$ and $K_{2}\left(\alpha \right)$ can be estimated by the quantiles α/2-th and 1−α/2-th of the empirical distribution of ${\bar{E}}_{n}^{w}J\left(X\right)$, respectively. In Table 1, we obtain the values of $K_{1}\left(\alpha \right)$ and $K_{2}\left(\alpha \right)$, for some sizes of sample *n*.

### 6.2. Power of the Test

In this part, the power of the proposed test statistic is compared with some others. These competing approaches are the one-sample Kolmogorov–Smirnov, [31,32]. To compute the *p*-values of these tests, a package called “uniftest” in R software version 4.0.5 was used. The results of the test statistics of our proposed ${\bar{E}}_{n}^{w}J\left(X\right)$, the Kolmogorov–Smirnov, Quesenberry and Miller, and the Frosini are symbolically shown by WCPJ, K-S, Q-M, and FRO, respectively.

To compute the power of the tests, a random sample, which assumed all possible values in the interval (0,1), was generated from the non-standard uniform distribution, such as beta or Kumaraswamy distributions, see, for example [33], whose supports varied between 0 and 1. After that, the powers were estimated empirically. We considered the following alternative distributions to compute the tests’ power:(1)Beta distribution with pdf: $1/B\left(a,b\right)x^{a-1}{\left(1-x\right)}^{b-1}$:(i)Beta (1.5, 1.5)(ii)Beta (0.5, 0.3)(iii)Beta (10, 1);(2)Kumaraswamy distribution with cdf: $1-{\left(1-x^{a}\right)}^{b}$:(i)Kuma (0.5, 5)(ii)Kuma (0.5, 0.3)(iii)Kuma (10, 10);(3)Piecewise distribution function with cdf $F\left(x\right)=\left\{\begin{array}{cc} 0.5-2^{k-1}{\left(0.5-x\right)}^{k}; & 0\leq x\leq 0.5\\ 0.5+2^{k-1}{\left(x-0.5\right)}^{k}; & 0.5\leq x\leq 1 \end{array}\right.$(i)Piec (2)(ii)Piec (3.5)(iii)Piec (5)

The results are depicted in Figure 1 for the different values of the sample size *n* as 20,30,40, and 50.

Figure 1 shows that the power of our proposed test based on the WCPJ was comparable to that of others for the beta and Kumaraswamy distributions. Even in some cases for these distributions, its power was superior to the other tests. For third alternative distribution, the power of the test based on the WCPJ was weaker than that of the rest. However, as the sample size *n* became larger, its power improved, and the test learned to discriminate the observations arising from the standard uniform distribution from those generated from nonuniform distributions. This test statistic can be satisfactory with due attention to its simple form and the rich information content behind it. Note that the plots for comparing the powers of the proposed test statistics are not shown in Figure 1 for beta (10, 1), kuma (0.5, 5) and kuma (10, 10), because the powers of all the tests were equal to 1.

## 7. Conclusions

The use of the extropy measure and its generalizations have become widespread in all scientific fields. One updated generalization of this measure is known as weighted extropy. In this paper, we introduced a new measure of uncertainty, related to cumulative extropy, named weighted cumulative past extropy (WCPJ). The properties of the WCPJ and a number of results including inequalities and various bounds to the WCPJ were considered. Studies related to reliability theory were discussed. A topic that may attract the attention of researchers is the dynamic version of the extropy in the sense that the uncertainty of the system depends on time *t*. Further research should investigate the uncertainty measure based on the weighted dynamic cumulative past or residual extropy. As an application of the proposed method, the empirical WCPJ was proposed to estimate this new information measure, and a test statistic was provided for the problem of the goodness of fit test of the standard uniform distribution based on the proposed WCPJ. Several applications of extropy and its generalizations, such as in information theory, economics, communication theory, and physics, can be found in the literature. Here, we cite some references. Ref. [34] studied the stock market in OECD countries based on a generalization of extropy known as negative cumulative extropy. Ref. [35] applied another version of extropy known as the Tsallis extropy to a pattern recognition problem. Ref. [16] explored an application of a generalization of extropy known as the fractional Deng extropy to a problem of classification. Ref. [36] used some extropy measures for the problem of compressive sensing.

## Figures and Tables

**Figure 1 entropy-24-01444-f001:**
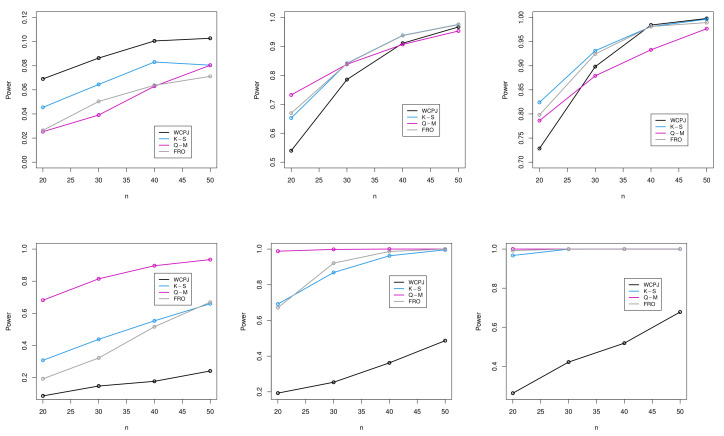
Power comparison of the WCPJ, K-S, Q-M, and FRO test statistics: above (**left**): beta (1.5, 1.5), (**middle**): beta (0.5, 0.3), and (**right**): kuma (0.5, 0.3); and below (**left**): piec (2), (**middle**): piec (3.5), and (**right**): piec (5) distributions.

**Table 1 entropy-24-01444-t001:** Values of $K_{1}\left(\alpha \right)$ and $K_{2}\left(\alpha \right)$ for α=0.05.

	*n*			
Cutoff Points	20	30	40	50
$K_{1}\left(\alpha \right)$	−0.1463	−0.1446	−0.1429	−0.1405
$K_{2}\left(\alpha \right)$	−0.0668	−0.0785	−0.0861	−0.0910

## Data Availability

Not applicable.

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
