# Peer review of "Weighted Cumulative Past Extropy and Its Inference"

_entropy, 2022, doi:10.3390/e24101444_

Round 1

Reviewer 1 Report

See the attached review report

Author Response

In attachment our reply. Thank you for your useful suggestions.

Reviewer 2 Report

Read carefully before final publication.  Please see attachment.

Author Response

Thank you for your useful suggestions

Reviewer 3 Report

The article ''Weighted Cumulative Past Extropy and its Inference'' presents a measure of uncertainty based on the concepts of empirical weighted cumulative past extropy (WCPJ) to deal with hypothesis testing issues. The topic is of interest to data scientists who work with the estimation of parameters from observed data sets. The manuscript is well written, and the objectives are achieved and enough presented. However, although not mandatory, graphic illustrations are desirable in order to help readers visualize the main results. In addition, perspectives for the development of the present methodology and possible applications are desirable in section Conclusion. The above points must be addressed before the manuscript can be considered for publication.

Author Response

Thank you for useful suggestions

Round 2

Reviewer 1 Report

The authors have revised the paper carefully based on my previous comments. I recommend its publication at the present form.